# Liver Tumor Markers, HALP Score, and NLR: Simple, Cost-Effective, Easily Accessible Indexes for Predicting Prognosis in ICC Patients after Surgery

**DOI:** 10.3390/jpm12122041

**Published:** 2022-12-09

**Authors:** Deyao Zhang, Huilan Zeng, Yangxun Pan, Yumo Zhao, Xin Wang, Jinbin Chen, Juncheng Wang, Yaojun Zhang, Zhongguo Zhou, Li Xu, Minshan Chen, Dandan Hu

**Affiliations:** 1Collaborative Innovation Center for Cancer Medicine, State Key Laboratory of Oncology in South China, Sun Yat-Sen University Cancer Center, Guangzhou 510060, China; 2Department of Liver Surgery, Sun Yat-Sen University Cancer Center, Guangzhou 510060, China; 3Department of Nuclear Medicine, Sun Yat-Sen University Cancer Center, Guangzhou 510060, China

**Keywords:** des-gamma-carboxyprothrombin, carbohydrate antigen 19-9, carcinoembryonic antigen, HALP score, NLR, PLR, prognosis

## Abstract

Introduction: To investigate the prognostic significance of liver tumor markers, the hemoglobin, albumin, lymphocyte, and platelet (HALP) score; neutrophil-to-lymphocyte ratio (NLR); and platelet-to-lymphocyte ratio (PLR), for predicting the specific site of recurrence or metastasis after surgery in patients with intrahepatic cholangiocarcinoma (ICC). Methods: In total, 162 patients with pathologically proven ICC who underwent curative surgery at Sun Yat-sen University Cancer Center between April 2016 and April 2020 were analyzed. Clinicopathological characteristics were collected retrospectively. The Kaplan–Meier method was used to analyze the overall survival (OS) and recurrence-free survival (RFS). Significant clinical factors were examined by univariate analysis and multivariate analysis and analyzed by receiver operating characteristic (ROC) curve analysis. Results: The cutoff values for the HALP score, NLR, and PLR were determined to be 43.63, 3.73, and 76.51, respectively, using the surv_cutpoint function of survminer using RFS as the target variable. In multivariate analysis, vascular invasion, pathology nerve tract invasion, and carbohydrate antigen 19-9 (CA19-9) levels were independent prognostic factors of OS, whereas the tumor number, pathology microvascular invasion, pathology differentiation, CA19-9 levels, and NLR were independent prognostic factors of RFS. For the whole recurrence analysis, the carcinoembryonic antigen (CEA) index exhibited the largest ROC curve area of all (AUC = 0.590), and the alpha-fetoprotein (AFP) index exhibited the smallest ROC curve area (AUC = 0.530). The HALP score exhibited the largest ROC curve area of all in predicting intrahepatic recurrence (AUC = 0.588), the NLR showed the best predictive value in predicting lymph node metastasis (AUC = 0.703), and the AUC of the CA19-9 index was the largest of all variables in predicting distant metastasis (AUC = 0.619). Conclusions: Our study showed that CA19-9, CEA, HALP score, and NLR are easily accessible, reliable, cost-effective indexes for predicting the specific site of recurrence or metastasis after surgery in ICC patients. Patients with high HALP scores and NLR have a higher risk of intrahepatic and lymph node metastasis recurrence.

## 1. Introduction

Intrahepatic cholangiocarcinoma (ICC) is the second-most common primary liver cancer, with an elevated incidence across the world [1]. Surgical resection is currently the only curative treatment strategy. Due to the high malignancy degree with insidious onset, invasive and fast-growing nature, and high rate of lymph node metastasis, most ICC patients are initially diagnosed late and lose the chance of surgical resection. Furthermore, the heterogeneity of ICC renders the early detection of postoperative tumor recurrence a major challenge. About 50% of patients experience recurrence within 2 years after curative resection, and five-year overall survival (OS) ranges from 21% to 35% [2,3,4]. The prognosis of ICC is affected by many factors, such as surgical quality, lymph node metastasis, and postoperative pathological features. Several studies have also reported that systemic nutritional and inflammation status play important roles in the development of cancer [5]. In addition, pretreatment serum tumor markers have important reference values for treatment planning and prognosis prediction.

Tumor markers are essential in the screening, early diagnosis, and differential diagnosis of cancer. The National Comprehensive Cancer Network (NCCN) guideline for ICC recommends laboratory tests of serum levels of alpha-fetoprotein (AFP), carbohydrate antigen 19-9 (CA19-9), and carcinoembryonic antigen (CEA). AFP has been widely accepted as a tumor marker for hepatocellular carcinoma in population-based screening and surveillance for many years. Approximately 20% of ICC patients are AFP-positive, and these patients are often misdiagnosed as hepatocellular carcinoma (HCC) before pathological diagnosis [6]. Therefore, a more sensitive and specific tumor marker is needed for diagnosis and prognosis. Accurate clinical classification is the basis of disease diagnosis and treatment. Currently, the most widely used staging system for ICC relies on the Tumor Node Metastasis (TNM) system designed by the American Joint Committee on Cancer (AJCC) [7]. Although the CA19-9 expression levels were not included in the AJCC TNM staging system, they were noted to be an independent prognosis factor in the manual. Mavros et al. confirmed that 57.5% of 429 ICC displayed preoperative CA19-9 levels (>35 U/mL) [2]. This study further confirmed that the expression level of CA19-9 was associated with liver cirrhosis and lymph node metastasis and was positively related to the chance of tumor recurrence. Therefore, CA19-9 is the most widely used tumor marker in the diagnosis and surveillance of ICC. However, a recent meta-analysis suggested that the overall pooled sensitivity of CA19-9 in diagnosing cholangiocarcinoma was 0.72 (0.70–0.75), and specificity was 0.84 (0.82–0.85) [8]. Other related studies have reported that CA19-9 has a wide variation in sensitivity (50–90%) and specificity (54–98%) [9,10]. Because the concentrations of CA19-9 could rise in patients with benign inflammation as well as in other malignancies [11], the single tumor marker of CA19-9 has a high false-negative and high false-positive rate and in the diagnosis of ICC, which cannot achieve the expected effect in clinical application. Thus, additional biomarkers are needed to improve the sensitivity and specificity of ICC diagnoses. CEA, which is known as an important tumor marker in diagnosing and predicting prognosis in many gastrointestinal tumors [12], has also been reported as an independent prognosis factor for cholangiocarcinoma, including ICC [3]. The sensitivity and specificity for CEA values were 53–84% and 50–79% [13,14]. CEA levels have also been reported to be effectively supplementary to CA19-9 levels in predicting survival outcomes for ICC patients after surgery, especially in patients with normal preoperative CA19-9 levels [15]. Furthermore, prothrombin induced by vitamin K absence-II (PIVKA-II), also known as des-gamma-carboxyprothrombin (DCP), has been proven to be not only a specific tumor marker but also a predictor of prognosis for hepatocellular carcinoma (HCC) [16]. According to our previous study, elevated levels of PIVKA-II in HCC patients stand for more aggressive tumor biology with a larger tumor size, increased percentage of vascular invasion, and worse prognosis [17]. We have noticed in clinical practice that positive PIVKA-II is not a rare phenomenon, but the application of PIVKA-II in ICC patients has never been reported in previous publications. As one of the critical tumor markers for liver cancer, whether PIVKA-II can provide useful information on the biological behavior of ICC remains unknown.

If all tumor markers of an ICC patient are negative, searching for some clinical indicators, such as hematological indexes, to predict the survival outcomes of ICC patients is meaningful. Hemoglobin, albumin, lymphocytes, neutrophils, and platelets have been reported to play crucial roles in tumorigenesis and development, but their mechanism is very complex. Elevated hemoglobin, albumin, and lymphocytes may be favorable prognostic factors, while elevated neutrophils and platelets may be unfavorable prognostic factors. Using a single index to predict prognosis could be affected by many biased factors, so combining multiple indexes could reduce the error. Recent studies have identified a variety of inflammatory or nutritional prognosis indexes, including hemoglobin, albumin, lymphocyte, and platelet (HALP) score [18]; neutrophil-to-lymphocyte ratio (NLR) [19]; platelet-to-lymphocyte ratio (PLR) [20]; and so on. However, until now, there has been very little research on how these parameters affect the prognosis of ICC patients.

Therefore, it is informative to inspect the values of liver tumor markers, HALP score, PLR, and NLR in patients from China in a large sample size to predict the prognosis of ICC patients and guide personalized treatment.

## 2. Materials and Methods 

### 2.1. Patients

A total of 162 patients with pathologically proven ICC who underwent radical surgery at Sun Yat-sen University Cancer Center between April 2016 and April 2020 were enrolled in our study.

The exclusion criteria included (1) pathologically confirmed HCC or combined HCC and ICC, (2) other synchronous malignancies, (3) lack of pretreatment tumor marker data, (4) a post-treatment survival time of less than 1 month, (5) lack of a follow-up assessment, and (6) liver function Child–Pugh C. 

We collected clinicopathological data from our hospital information system, and lab test data were collected within three days before surgery. AJCC TNM staging, Child–Pugh classifications, and Barcelona Clinic Liver Cancer (BCLC) staging were calculated accordingly.

### 2.2. Follow-Up and Data

Patients were followed-up every three months up to two years and every six months until year five. Patients needed to be regularly followed-up with radiological examinations, such as computerized tomography (CT) scans or magnetic resonance imaging (MRI) and chest X-rays or CT. Laboratory tests included blood routines, liver function, and liver tumor markers. The cutoff values of AFP, CA19-9, CEA, and PIVKA-II were set to be 25 ng/mL, 35 U/mL, 5 ng/mL, and 40 mAU/mL, respectively, according to previous studies. If the values of these tumor markers were above the cutoff value, they were considered positive; HALP [hemoglobin (g/L) × albumin (g/L) × lymphocyte (/L)/platelet (/L)] and NLR [neutrophil (/L)/lymphocyte (/L)]; PLR [platelet (/L)/lymphocyte (/L)]. The HALP score, PLR, and NLR optimal cut-off point to separate continuous variables were identified using the surv_cutpoint function of survminer using recurrence-free survival (RFS) as the target variable. Appendix A shows that the optimal thresholds of HALP score, PLR, and NLR for predicting ICC recurrence were 43.63, 76.51, and 3.73, respectively. OS was calculated from the date of surgery to the date of death or the last follow-up time. RFS was calculated from the date when the tumor recurrence was diagnosed or the last follow-up time. According to the relapse sites after surgery, recurrence was divided into intrahepatic tumor recurrence, lymph node metastasis, and distant metastasis. Correspondingly, intrahepatic tumor RFS, lymph node metastasis RFS, and distant metastasis RFS were calculated.

### 2.3. Statistical Analysis

All data were presented as mean 95% confidence intervals (CIs). The Pearson chi-square test was used to analyze categorical variables, and a two-tailed *t*-test was used to analyze the continuous variables. The Kaplan–Meier technique was employed to estimate the OS and RFS, and the log-rank test was applied to compare them. Significant clinical factors were examined by univariate analysis and multivariate analysis and analyzed by receiver operating characteristic (ROC) curve analyses.

We performed all statistical analyses with SPSS software version 26.0 (IBM Corporation, Armonk, NY, USA) and R version 3.5.1 (R Foundation for Statistical Computing, Vienna, Austria, http://www.r-project.org (accessed on 1 September 2021)). 

## 3. Results

### 3.1. Clinicopathological Characteristics 

Table 1 outlines the clinicopathological features of the 162 enrolled patients. A total of 100 males (61.7%) with a median age of 56.1 years (range, 32–76 years) and 86 females (37.4%) with a median age of 57.4 years (range, 40–77 years) were included. There were 73 (45.1%) patients whose serum CA19-9 levels were positive (>35 U/mL), which was the highest among the four liver tumor markers. The positive rates of AFP, CEA, and PIVKA-II were 12.3%, 22.2%, and 20.4%, respectively. The patients with negative tumor markers accounted for 35.2%, and patients with single-positive and double- or more positive tumor markers accounted for 34.0% and 30.8%, respectively. The proportion of HBsAg-positive ICC patients was 52.5%. (Table 1). 

We then analyzed the correlation between different liver tumor markers and clinicopathological features. Our study found that the AFP level was significantly associated with gender and hepatitis B virus infection status (*p* < 0.05; Appendix A). However, it did not correlate with OS and RFS (*p* > 0.05). Elevated CA19-9 was associated with decreased albumin levels, more advanced BCLC staging and AJCC staging, and the presence of lymph node metastasis (*p* < 0.05). Similar to CA19-9, the serum levels of CEA showed statistical significance with total bilirubin levels, BCLC staging, AJCC staging, tumor numbers, OS, RFS, and recurrence (*p* < 0.05). An increased level of PIVKA-II was related to more advanced BCLC staging and AJCC staging and fewer tumor numbers (*p* < 0.05). These four liver tumor markers did not exhibit a specific ability in predicting relapse sites (*p* > 0.05). However, patients in the HALP ≥ 43.63 group showed a greater proportion of intrahepatic recurrence than patients in the HALP < 43.63 group (43.0% vs. 25.8%, *p* = 0.027). More patients in the NLR ≥ 3.73 group had a higher rate of lymph node metastasis recurrence than patients in the NLR < 3.73 group.

### 3.2. Overall Survival and Recurrence-Free Survival Analysis

We also analyzed the relationship between four kinds of liver tumor markers and the prognosis of ICC patients (Figure 1A–D and Figure 2A–D). Patients with lower pretreatment CA19-9 and CEA levels had a better longer OS and RFS time (*p* < 0.05). However, pretreated AFP and PIVKA-II levels were not significant in the OS of the ICC patients. In addition, hepatitis B or C history was not associated with prognosis (Appendix A). Furthermore, ICC with microvascular invasion (MVI) may have worse surgical outcomes with regard to RFS but not OS (Appendix A). Differing from MVI, nerve tract invasion was significantly associated with OS but did not affect RFS (Appendix A). Survival analysis also indicated that BCLC staging systems had predictive value for the prognosis of ICC (Appendix A). Patients with positive lymph nodes showed poorer prognoses than those with negative lymph nodes (Appendix A).

Then, survival analysis for the intrahepatic tumor RFS (Appendix A), lymph node metastasis RFS (Appendix A), and distant metastasis RFS (Appendix A) was calculated correspondingly. Intrahepatic recurrence is most common after liver resection. However, a single liver tumor marker or a different combination of them (AFP + PIVKA-II, AFP + CEA, AFP + CA19-9, PIVKA-II + CEA, PIVKA-II + CA19-9, and CEA + CA19-9) did not indicate reliable predictive power of intrahepatic recurrence (*p* > 0.05) (Appendix A). Likewise, this situation also occurred in predicting lymph node metastasis recurrence (*p* > 0.05) (Appendix A). Appendix A shows that a single CA19-9 or CEA has the potential predictive power of distant metastasis after surgery (*p* < 0.05). A Kaplan–Meier analysis was also performed to determine the relationship between the HALP score (Figure 3), PLR (Figure 4), NLR (Figure 5), and RFS. Survival analysis revealed that the HALP score could have the potential predictive ability of intrahepatic recurrence (*p* = 0.053) (Figure 3). The predictive prognosis ability of PLR did not reach statistical significance (*p* > 0.05) (Figure 4). Moreover, we found that the NLR generated good performance in predicting the survival outcome in intrahepatic recurrence (*p* = 0.026) and lymph node metastasis (*p* = 0.0004) (Figure 5).

### 3.3. Univariate and Multivariable Cox Regression Analyses in Cohorts

The univariate analyses showed that vascular invasion, pathology nerve tract invasion, CA19-9, CEA, and NLR levels were associated with OS. Tumor number, tumor size, vascular invasion, pathological MVI, pathology differentiation, and CA19-9, CEA, and NLR levels were associated with RFS. In the multivariate study, we demonstrated that vascular invasion, pathology nerve tract invasion, and CA19-9 levels remained independent prognostic factors of OS, and tumor number, MVI, pathology differentiation, and CA19-9 and NLR levels were still independent prognostic factors of RFS (Table 2 and Table 3).

To verify the value of liver tumor markers, HALP score, PLR, and NLR on the prognosis of ICC patients, we analyzed the relationship between recurrent sites and the clinicopathological parameters, and the results of univariate analysis are summarized in Table 4. Tumor number, tumor size, vascular invasion, pathology MVI, pathology differentiation degree, and NLR were significantly associated with intrahepatic tumor recurrence on univariate analysis. Notably, the NLR was the only factor significantly associated with lymph node metastasis recurrence (hazard ratio (HR): 7.757, 95% confidence interval (CI0: 2.031–29.619; *p* = 0.003). In addition, vascular invasion, CEA, and a history of cholelithiasis were associated with a higher rate of distant metastasis recurrence. 

### 3.4. Comparison of AFP, PIVKA-II, CEA, CA19-9, HALP Score, PLR, and NLR 

In addition, the predictive accuracy of seven variables (AFP, PIVKA-II, CEA, CA19-9, HALP score, PLR, and NLR), which were readily available before the operation, was analyzed using ROC curves (Figure 6). For the whole recurrence analysis, the CEA index exhibited the largest ROC curve area of all (AUC = 0.590), and the AFP index exhibited the smallest ROC curve area (AUC = 0.530) (Figure 6A). Surprisingly, the HALP score exhibited the largest ROC curve area of all in predicting intrahepatic recurrence (AUC = 0.588) (Figure 6B), and the NLR showed the best predictive value in predicting lymph node metastasis recurrence (AUC = 0.703) (Figure 6C). Last but not least, the AUC of the CA19-9 index was the largest of all variables in predicting distant metastasis recurrence (AUC = 0.619) (Figure 6C).

## 4. Discussion

ICC is one of the highest degrees of malignancies, with rising incidence and mortality rates around the world [21,22]. Despite continued advances in new treatment options, the prognosis of ICC patients who cannot undergo curative resection is still unsatisfactory. In general, early diagnosis is necessary for beneficial outcomes, and tumor markers have a noticeable effect on the diagnosis and follow-up of cancer patients. The most commonly used tumor markers for liver cancers include CA19-9, CEA, AFP, and PIVKA-II. Here, we investigated the diagnostic and prognostic value of these four liver tumor markers in 162 ICC patients. Although PIVKA-II has a comparable positive rate with CEA in ICC patients, this is the first study to investigate the application of PIVKA-II in ICC patients.

In this study, the positivity rate of serum CA19-9 levels was about 45.1%, which agrees well with previous studies [2]. Univariate and multivariate Cox proportional hazard regression analyses showed that pretreated serum CA19-9 levels were independently prognostic. Patients with lower CA19-9 levels had better long-term OS and RFS (*p* < 0.05). According to previous research, preoperative CEA levels were complementary to CA19-9 levels in predicting prognosis in patients with resectable ICC [15], and univariate analysis in our study also revealed that evaluated serum CEA levels were related to poor prognosis. These two tumor markers were previously confirmed and commonly applied in the diagnosis of ICC. However, the NCCN guideline for ICC recommends that the checkup should include CA19-9, CEA, and AFP. The inclusion of AFP was considered to help in the differential diagnosis of the suspected liver mass. Both AFP and PIVKA-II have been used for the early screening and diagnosis of HCC as a routine [23,24], but the evidence regarding ICC is sparse. In our study, the positive rate of PIVKA-II for ICC was about 20.4%, which was close to 22.2% for CEA. The elevated levels of PIVKA-II were positively correlated with gender ratio, higher rate of HBV infection, more advanced BCLC staging, and AJCC staging (Appendix A), and these are similar clinicopathologic features that were reported in PIVKA-II-positive HCC [17]. PIVKA-II was reported to be related to abnormal vitamin K metabolism and defects in the post-translational carboxylation of the prothrombin precursor [25,26,27]. The serum levels of PIVKA-II are elevated in disorders of hepatic function and are known as an important serum marker of HCC [28]. However, PIVKA-II is not taken as a routine test in ICC. In the latest National Comprehensive Cancer Network ALL guidelines (NCCN Clinical Practice Guidelines in Oncology: Hepatobiliary Cancers), CA19-9, CEA, and AFP were recommended as tests to diagnose ICC [29], while serum PIVKA-II levels were not mentioned. In our study, patients with HBV infection accounted for 66.7% of PIVKA-II-positive patients. Previous studies showed that the clinicopathological similarities between HBV-associated ICC and HBV-associated HCC were numerous [30], and patients with HBV-ICC achieved better outcomes than those without HBV infection [31]. Thus, the features of PIVKA-II-positive patients can be concluded as a more advanced stage with HBV infection, which has opposite effects on the prognosis of ICC. This may be the reason why PIVKA-II-positive patients tended to have a higher degree of malignancy but did not show a worse prognosis than PIVKA-II-negative patients.

It is important to find valid indexes to predict the specific site of recurrence or metastasis after surgery in ICC patients. Patients with only intrahepatic recurrence could have a second chance of surgery, while patients with lymph node metastasis or distant metastasis may lose the chance of receiving a second operation and receive chemotherapy or immunotherapy. Therefore, we also examined the relationship between different recurrence patterns and liver tumor markers. Regrettably, survival analysis for intrahepatic tumor recurrence and lymph node metastasis recurrence showed that single-use or combined use of these four liver tumor markers did not show exhibit reliable prognostic power. In predicting distant metastasis recurrence, CEA did have important prognostic significance in ICC patients (HR: 2.920, 95%CI 1.033–8.255, *p* = 0.043). Thus, we need to find some novel prognostic indicators for ICC patients, especially for those whose liver tumor markers were all negative.

Tumor is a chronic wasting disease characterized by nutritional, inflammation, and metabolic disorders [32]. An increasing number of researchers have noticed the effect of the inflammatory response and the nutritional status of patients on the prognosis of tumors [33]. The HALP score, PLR, and NLR, which combine convenient and economical blood indicators, have been proven to be effective prognostic factors for multiple tumors [18,19]. However, few related studies have been carried out on ICC patients. Therefore, it is meaningful to study the correlation between liver tumor markers, HALP score, PLR, NLR, and clinicopathological characteristics and prognosis of ICC.

In our study, the HALP score, NLR, and PLR provided a good complement to liver tumor markers. Univariate survival analysis showed that a lower HALP score (HALP score < 43.63) and lower NLR (NLR < 3.73) were associated with lower intrahepatic recurrence risk. In addition, NLR was the only potential index to predict the risk of lymph node metastasis recurrence postoperatively (HR: 7.757, 95% CI 2.031–29.619, *p* = 0.003). Subjects with a history of cholelithiasis were also more likely to experience distant metastasis recurrence (HR: 3.106, 95% CI 1.105–8.729, *p* = 0.032), which is consistent with previous research. Cai et al. [34] made a meta-analysis to indicate that bile duct stones, including choledocholithiasis, were an important risk factor for ICC (odds ratio (OR): 11.79, 95% CI 4.17–33.35). Then, the performance was analyzed using ROC curves and areas under ROC curves (AUC). The HALP score exhibited the largest ROC curve area of all in predicting intrahepatic recurrence (AUC = 0.588), and NLR showed the best predictive value in predicting lymph node metastasis (AUC = 0.703). Last but not least, the AUC of the CA19-9 index was the largest of all variables in predicting distant metastasis (AUC = 0.619). Thus, the combination of nutritional and inflammatory indexes could represent a valuable addition to liver tumor markers and the appraisal of recurrence risk before the operation for tailoring individualized treatment.

Although this study is the first to combine the HALP score, PLR, NLR, and liver tumor markers as independent predictive factors in ICC patients, the retrospective single-center study design is a limiting factor of our study. In addition, the cutoff of the HALP score, NLR, and PLR failed to reach a common consensus value due to the limited number of patients, which could affect the accuracy of the results. Moreover, with the emerging studies relating the next-generation sequencing (NGS) results and treatment options, adding genetic profile into consideration is definitely the future trend. We plan to enlarge our cohort and design a multicenter study to further analyze these scores and liver tumor markers in predicting ICC patients’ prognoses.

## 5. Conclusions

To conclude, CA19-9, CEA, HALP score, and NLR are easily accessible, reliable, and cost-effective indexes for predicting the specific site of recurrence or metastasis after surgery in ICC patients. Patients with high HALP scores and NLR have a higher risk of intrahepatic and lymph node metastasis recurrence.

## Figures and Tables

**Figure 1 jpm-12-02041-f001:**
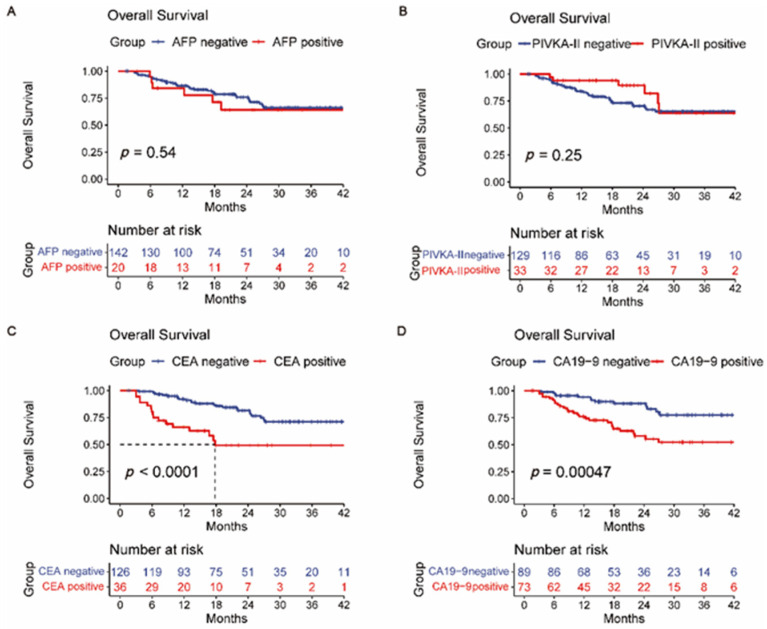
Overall survival (OS) curves of grouped intrahepatic cholangiocarcinoma patients based on (**A**) AFP, (**B**) PIVKA-II, (**C**) CEA, and (**D**) CA19-9.

**Figure 2 jpm-12-02041-f002:**
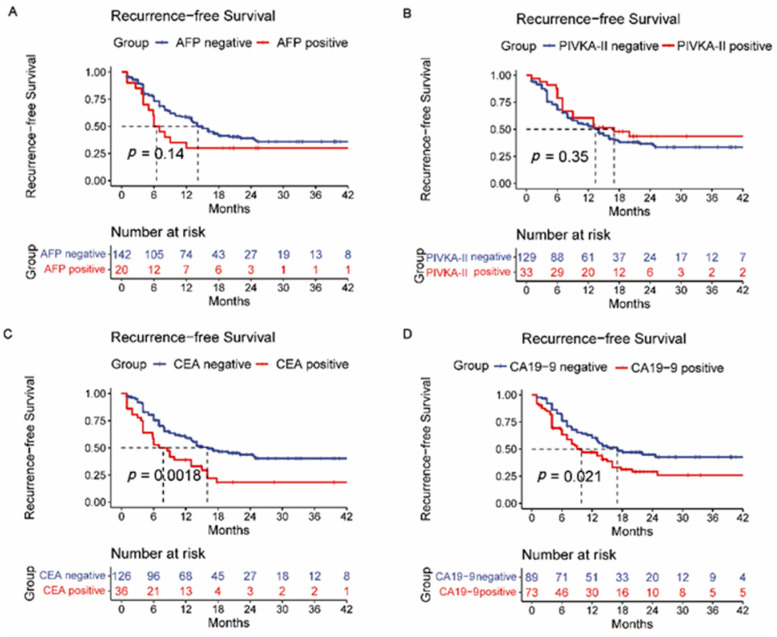
Recurrence-free survival (RFS) curves of grouped intrahepatic cholangiocarcinoma patients based on (**A**) AFP, (**B**) PIVKA-II, (**C**) CEA, and (**D**) CA19-9.

**Figure 3 jpm-12-02041-f003:**
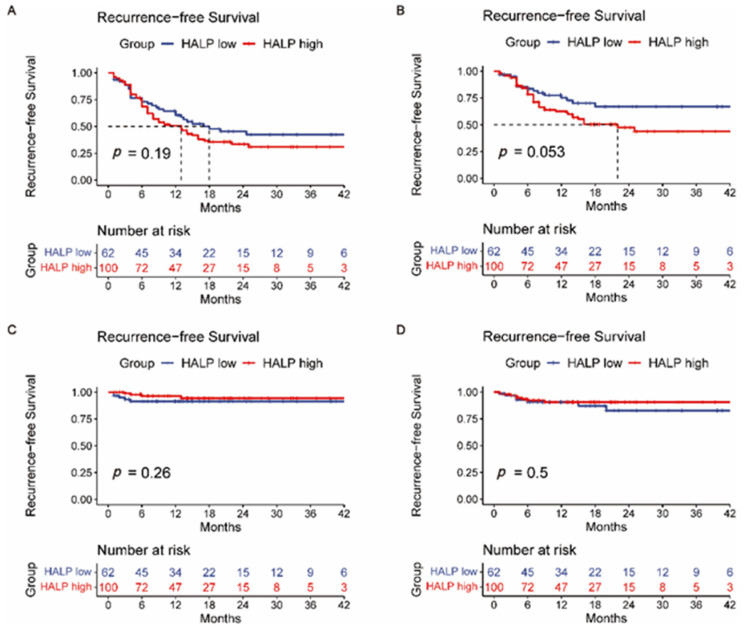
The value of HALP score in predicting (**A**) recurrence-free survival, (**B**) intrahepatic recurrence, (**C**) lymph node metastasis, and (**D**) distant metastasis recurrence.

**Figure 4 jpm-12-02041-f004:**
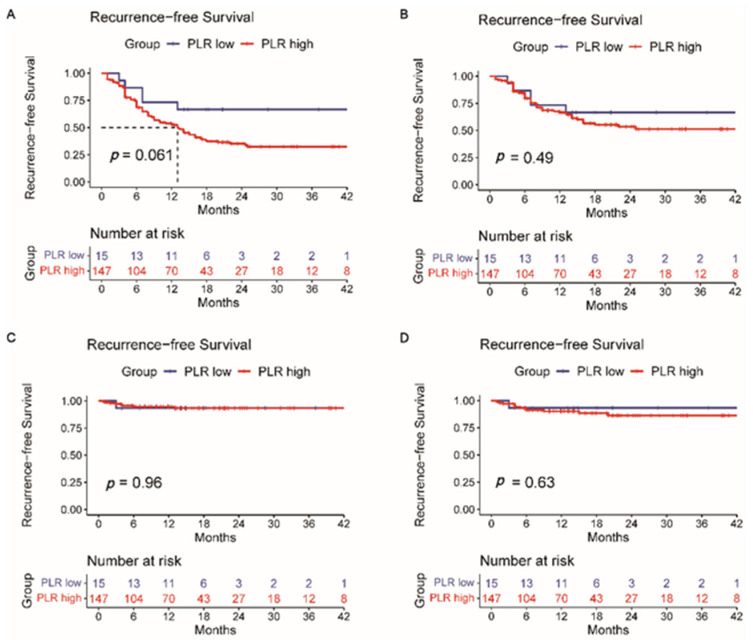
The value of PLR in predicting (**A**) recurrence-free survival, (**B**) intrahepatic recurrence, (**C**) lymph node metastasis, and (**D**) distant metastasis recurrence.

**Figure 5 jpm-12-02041-f005:**
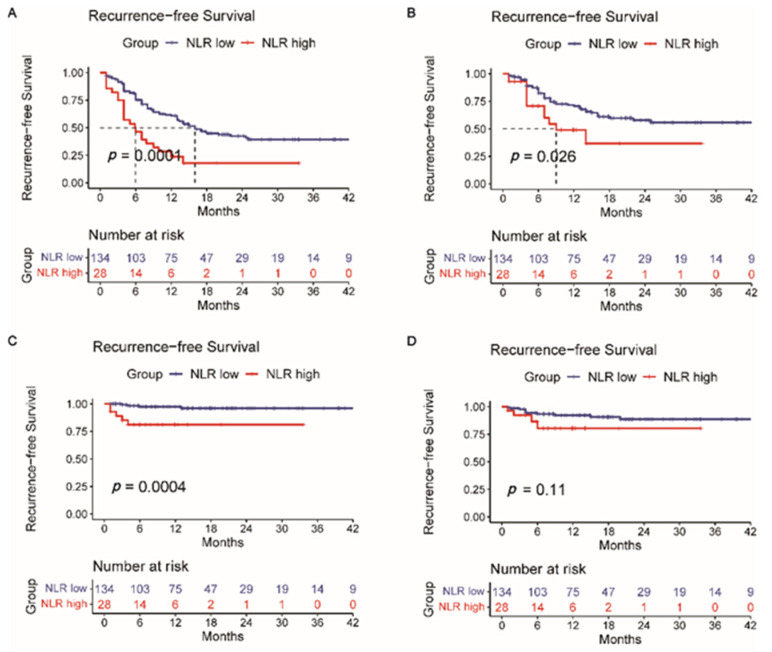
The value of NLR in predicting (**A**) recurrence-free survival, (**B**) intrahepatic recurrence, (**C**) lymph node metastasis, and (**D**) distant metastasis recurrence.

**Figure 6 jpm-12-02041-f006:**
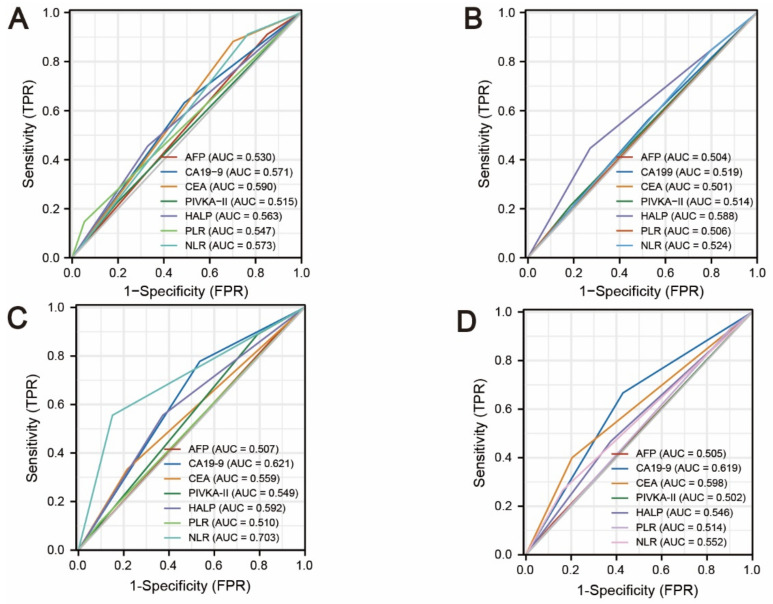
The area under the curve (AUC) for HALP, PLR, NLR, AFP, PIVKA-II, CEA, and CA19-9; (**A**) AUC of recurrence-free survival; (**B**) AUC of intrahepatic recurrence; (**C**) AUC of lymph node metastasis; and (**D**) AUC of distant metastasis recurrence.

**Table 1 jpm-12-02041-t001:** Demographical characteristics and clinical data of the patients.

Patient Characteristics		Number	Percentage
Age	≥56	91	56.2%
	<56	71	43.8%
Gender	Male	100	61.7%
	Female	62	37.4%
Tumor numbers	Single	108	66.7%
	Multiple	54	33.3%
Tumor size	≤2.0 cm	5	3.1%
	2.1–5.0 cm	70	43.2%
	>5.0 cm	87	53.7%
Hepatis B surface antigen	Positive	85	52.5%
	Negative	77	47.5%
Hepatitis C virus antibody	Positive	1	0.6%
	Negative	161	99.4%
Platelet count (×10^9^/L)	242.8 ± 87.1		
Albumin (mg/dL)	43.6 ± 3.7		
Alanine aminotransferase (U/L)	35.4 ± 33.6		
Aspartate aminotransferase (U/L)	32.6 ± 27.2		
Total bilirubin (umol/L)	13.9 ± 19.9		
Prothrombin time (s)	12.0 ± 4.9		
T stage	1	89	54.9%
	2	58	35.8%
	3	1	0.6%
	4	14	8.6%
N stage	0	135	83.3%
	1	27	16.7%
M stage	0	153	94.4%
	1	9	5.6%
AJCC TNM staging system	I	82	50.6%
	II	41	25.3%
	III	27	16.7%
	IV	12	7.4%
The number of elevated pretreatment serum liver tumor markers	0	57	35.2%
	1	55	34.0%
	≥2	50	30.8%
Vascular invasion	Yes	33	20.4%
	No	129	79.6%
Microvascular invasion	Yes	43	26.5%
	No	117	72.2%
	Unknown	2	1.2%
Nerve tract invasion	Yes	30	18.5%
	No	130	80.2%
	Unknown	2	1.2%
Tumor differentiations	Well	1	0.6%
	Well-moderate	1	0.6%
	Moderate	45	27.8%
	Moderate to poor	83	51.2%
	Poor	21	13.0%
	Unknown	11	6.8%
Alpha-fetoprotein (25 ng/mL)	421.9 ± 3664.4		
Cancer antigen 199 (35 U/mL)	715.4 ± 2306.8		
Carcinoembryonic antigen (5 ng/mL)	6.76 ± 19.6		
Protein induced by vitamin K absence or Antagonist-II (40 mAU/mL)	400.7 ± 4112.3		
Alpha-fetoprotein (25 ng/mL)	Elevated	20	12.3%
	Normal	142	87.7%
Cancer antigen 199 (35 U/mL)	Elevated	73	45.1%
	Normal	89	54.9%
Carcinoembryonic antigen (5 ng/mL)	Elevated	36	22.2%
	Normal	126	77.8%
Protein induced by vitamin K absence or Antagonist-II (40 mAU/mL)	Elevated	33	20.4%
	Normal	129	79.6%
Hemoglobin, albumin, lymphocyte, and platelet score	54.3 ± 30.1		
Neutrophil-to-lymphocyte ratio	2.9 ± 2.8		
Platelet-to-lymphocyte ratio	140.0 ± 66.6		
History of cholelithiasis	Yes	30	18.5%
	No	132	81.5%
Child–Pugh classifications	A	160	98.8%
	B	2	1.2%
Barcelona Clinic Liver Cancer staging	A	100	61.7%
	B	20	12.3%
	C	42	25.9%
Recurrence	Yes	94	58.0
	NO	68	42.0%
Intrahepatic tumor recurrence	Yes	59	36.4%
	NO	103	63.6%
Lymph node metastasis	Yes	9	5.6%
	NO	153	94.4%
Distant metastasis	Yes	15	9.3%
	NO	147	90.7%

**Table 2 jpm-12-02041-t002:** Univariate and multivariate analysis for OS in patients with ICC.

Variables	Overall Survival Univariate Analysis			Overall Survival Multivariate Analysis		
	HR	95% CI	*p*-Value	HR	95% CI	*p*-Value
Age (≥56 vs. <56)	1.725	0.916–3.248	0.091			
Gender (male vs. female)	0.977	0.718–1.330	0.883			
Cirrhosis (yes vs. no)	1.146	0.614–2.138	0.669			
Tumor number (multiple vs. single)	1.757	0.953–3.239	0.071			
Tumor size (>5 cm vs. ≤5 cm)	1.800	0.963–3.364	0.065			
Vascular invasion	2.992	1.615–5.543	<0.001 *	2.347	1.216–4.530	0.011 *
Pathology microvascular invasion	1.311	0.681–2.523	0.418			
Pathology nerve tract invasion	2.846	1.490–5.437	0.002 *	3.018	1.536–5.927	0.001 *
Pathology differentiation degree (Moderate to poor, poor vs. well, well to moderate, moderate)	1.483	0.719–3.056	0.286			
AFP (≥25 vs. <25)	1.236	0.521–2.935	0.631			
CA19-9 (≥35 vs. <35)	2.653	1.410–4.989	0.002 *	2.071	1.030–4.167	0.041 *
CEA (≥5 vs. <5)	3.384	1.818–6.300	<0.001 *	1.965	0.880–3.746	0.107
PIVKA-II (≥40 vs. <40)	0.463	0.182–1.180	0.107			
History of cholelithiasis	1.660	0.909–3.032	0.099			
Hepatitis B	0.920	0.502–1.687	0.788			
HALP (≥43.63 vs. <43.63)	1.258	0.662–2.391	0.483			
NLR (≥3.73 vs. <3.73)	0.475	0.238–0.946	0.034 *	1.463	0.691–3.098	0.320
PLR (≥76.51 vs. <76.51)	0.217	0.030–1.580	0.132			

Notes: * *p* < 0.05.

**Table 3 jpm-12-02041-t003:** Univariate and multivariate analysis for RFS in patients with ICC.

Variables	Recurrence-Free Survival Univariate Analysis			Recurrence-Free Survival Multivariate Analysis		
	HR	95% CI	*p*-Value	HR	95% CI	*p*-Value
Age (≥56 vs. <56)	1.088	0.723–1.636	0.686			
Gender (male vs. female)	1.172	0.948–1.449	0.142			
Cirrhosis (yes vs.no)	1.277	0.844–1.933	0.247			
Tumor number (multiple vs. single)	2.491	1.651–3.759	<0.001 *	2.060	1.250–3.397	0.005 *
Tumor size (>5 cm vs. ≤5 cm)	1.925	1.265–2.928	0.002 *	1.315	0.813–2.127	0.265
Vascular invasion	2.539	1.619–3.982	<0.001 *	1.561	0.931–2.617	0.091
Pathology microvascular invasion	1.980	1.290–3.040	0.002 *	1.800	1.121–2.892	0.015 *
Pathology nerve tract invasion	1.204	0.719–2.015	0.481			
Pathology differentiation degree (moderate to poor, poor vs. well, well to moderate, moderate)	1.976	1.187–3.290	0.009*	1.826	1.054–3.163	0.032 *
AFP (≥25 vs. <25)	1.514	0.858–2.675	0.153			
CA19-9 (≥35 vs. <35)	1.587	1.058–2.380	0.025 *	1.794	1.117–2.882	0.016 *
CEA (≥5 vs. <5)	1.972	1.264–3.074	0.003 *	1.037	0.609–1.766	0.893
PIVKA-II (≥40 vs. <40)	0.789	0.472–1.320	0.789			
History of cholelithiasis	1.326	0.809–2.173	0.263			
Hepatitis B	1.056	0.704–1.585	0.792			
HALP (≥43.63 vs. <43.63)	1.319	0.857–2.030	0.208			
NLR (≥3.73 vs. <3.73)	0.400	0.246–0.652	0.000 *	2.424	1.386–4.239	0.002 *
PLR (≥76.51 vs. <76.51)	0.439	0.178–1.082	0.074			

Notes: * *p* < 0.05.

**Table 4 jpm-12-02041-t004:** Univariate analysis for intrahepatic tumor recurrence, lymph node metastasis, and distant metastasis in patients with ICC.

	Intrahepatic Tumor Recurrence		Lymph Node Metastasis		Distant Metastasis	
	Univariate Analysis		Univariate Analysis		Univariate Analysis	
	Hazard Ratio (95% CI)	*p*-Value	Hazard Ratio (95% CI)	*p*-Value	Hazard Ratio (95% CI)	*p*-Value
Age (≥56 vs. <56)	1.320 (0.782–2.229)	0.298	1.635 (0.409–6.538)	0.487	0.549 (0.195–1.542)	0.255
Gender (male vs. female)	1.202 (0.918–1.574)	0.182	0.924 (0.478–1.783)	0.813	1.036 (0.617–1.738)	0.895
Cirrhosis (yes vs. no)	1.196 (0.711–2.012)	0.500	0.634 (0.170–2.361)	0.497	0.923 (0.335–2.548)	0.878
Tumor number (multiple vs. single)	2.461 (1.465–4.135)	0.001 *	1.852 (0.494–6.946)	0.361	1.667 (0.588–4.727)	0.336
Tumor size (>5 cm vs. ≤5 cm)	1.977 (1.162–3.364)	0.012 *	7.836 (0.977–62.857)	0.053	2.971 (0.938–9.408)	0.064
Vascular invasion	1.983 (1.094–3.591)	0.024 *	0.584 (0.073–4.682)	0.613	3.496 (1.227–9.966)	0.019 *
Pathology microvascular invasion	1.832 (1.063–3.158)	0.029 *	0.438 (0.054–3.562)	0.440	1.301 (0.406–4.170)	0.658
Pathology nerve tract invasion	0.794 (0.377–1.676)	0.546	0.600 (0.075–4.799)	0.630	0.354 (0.046–2.690)	0.315
Pathology differentiation degree (moderate to poor, poor vs. well, well to moderate, moderate)	1.971 (1.037–3.743)	0.038 *	3.919 (0.490–31.359)	0.198	2.911 (0.643–13.173)	0.165
AFP (≥25 vs. <25)	1.171 (0.531–2.579)	0.696	0.983 (0.123–7.875)	0.987	1.233 (0.278–5.472)	0.783
CA19-9 (≥35 vs. <35)	1.365 (0.818–2.277)	0.234	0.393 (0.082–1.894)	0.244	2.891 (0.986–8.473)	0.053
CEA (≥5 vs. <5)	1.302 (0.702–2.414)	0.403	2.012 (0.502–8.059)	0.323	2.920 (1.033–8.255)	0.043 *
PIVKA-II (≥40 vs. <40)	0.763 (0.962–1.141)	0.419	0.435 (0.054–3.480)	0.433	0.858 (0.242–3.042)	0.813
History of cholelithiasis	1.259 (0.668–2.347)	0.476	2.324 (0.581–9.296)	0.233	3.106 (1.105–8.729)	0.032 *
Hepatitis B	0.976 (0.585–1.627)	0.925	1.068 (0.287–3.980)	0.922	1.741 (0.595–5.099)	0.312
HALP (≥43.63 vs. <43.63)	1.732 (0.975–3.078)	0.061	0.478 (0.128–1.782)	0.272	0.703 (0.255–1.941)	0.703
NLR (≥3.73 vs. <3.73)	2.020 (1.061–3.847)	0.032 *	7.757 (2.031–29.619)	0.003 *	2.468 (0.769–7.921)	0.129
PLR (≥76.51 vs. <76.51)	1.375 (0.549–3.438)	0.497	0.944 (0.118–7.566)	0.957	1.639 (0.215–12.467)	0.633

Notes: * *p* < 0.05.

## Data Availability

Not applicable; anonymized data will be supplied upon request to the corresponding author.

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
