# Peer review of "Liver Tumor Markers, HALP Score, and NLR: Simple, Cost-Effective, Easily Accessible Indexes for Predicting Prognosis in ICC Patients after Surgery"

_jpm, 2022, doi:10.3390/jpm12122041_

Round 1

Reviewer 1 Report

The mauscript is interesting and well written. Several analysis have been conducted to indentify prognostic markers for intraheaptic cholangiocarcinoma. Unfortunately most of them didn't show significant predictive power.

I have some comments, that are the following:

1) The main limitation of the manuscript is that several results are reported but few of them are significant and it is difficult for the reader to get the message of the study.

2) The results should be focuesd on a main endpoint and just some secondary endpoints. In the current version in fact, a lot of data regarding  not only prognostic markers but also prognostic factors are reported.

2) There are too many survival curves. The most important ones, reporting the significant results on the main endpoints should be reported and commented in the manuscript, while the others can be reported in the supplemenetary material.

3) The serum markers investigated are  rarely expressed by the patients of the series. Which of them could be useful, alone or in combination with others?

4) The manuscript is too long and should be shortened.

Author Response

Comment 1: 1) The main limitation of the manuscript is that several results are reported but few of them are significant and it is difficult for the reader to get the message of the study.

Reply: Thank you very much for your comments and professional advice. Comparing to hepatocellular carcinoma (HCC), intrahepatic cholangiocarcinoma (ICC) has not been studied comprehensively. The clinical diagnostic standard and prognostic factors of ICC have not reached a consensus among practitioners. We designed this pilot study to investigate the post-operative prognosis of ICC patients. To our knowledge, this is the first study exploring the relationship between clinical traits, tumor markers including PIVKA-II, inflammatory indicators and survival of ICC patients. Many of the results are insignificant which cannot translate into clinical usage, but they reflect the nature of the disease. To improve the structure of our manuscript, we reduced the length of the article and put some figures in the supplemental materials to make it easier for readers to get the message of the study.

Comment 2: The results should be focuesd on a main endpoint and just some secondary endpoints. In the current version in fact, a lot of data regarding not only prognostic markers but also prognostic factors are reported.

Reply: In this research, we found that only CA19-9 in liver tumor markers was related to the RFS in ICC patients. However, the modes of recurrence in ICC patients often conclude intrahepatic recurrence, lymph node metastasis, and distant metastasis, and the treatment method is different after relapse. So, we made this research to investigate whether these single liver tumor marker or their combination could have the prognostic prediction ability in these different modes of recurrence. Regrettably, only in predicting distant metastasis, CA19-9 had a nice ability, while in predicting intrahepatic recurrence, and lymph node metastasis, liver tumor markers didn’t achieve the desired effect. So, we focused on the preoperative HALP score, NLR, and PLR, which had been reported to have predictive value in breast cancer (PMID: 35702128), small-cell lung cancer (PMID: 33004760), myeloma (PMID: 36053932), and so on. Surprisingly, HALP had a potentially predictive ability in predicting intrahepatic recurrence and NLR had a nice predictive value in predicting lymph node metastasis recurrence. And these are simple, cost-effective, easily accessible indexes.

Comment 3: 2) There are too many survival curves. The most important ones, reporting the significant results on the main endpoints should be reported and commented in the manuscript, while the others can be reported in the supplementary material.

Reply: Thank you very much for your thoughtful advice, we have rearranged the figures and replaced some figures in the supplemental materials to make it more clear and concise. The revised manuscript with changes accordingly is uploaded.

Comment 4: The serum markers investigated are rarely expressed by the patients of the series. Which of them could be useful, alone or in combination with others?

Reply: Thank you for your comment. Liver tumor markers (AFP, PIVKA-II, CEA, CA19-9) are usually used for screening diagnoses for suspicious lesions in the liver. In our previous, we made an analysis of 4792 patients and proved that AFP and PIVKA-II could be used as important prognostic markers in HCC patients (PMID: 34235104). CA19-9 was reported as a valid prognostic marker for ICC (PMID: 23358969), and CEA was also reported to be supplementary to CA19-9 in ICC patients  (PMID: 30210635). However, few reports focused on the value of PIVKA-II in ICC patients. That’s one of the objectives of this research. So in this research, we talked about single liver tumor marker and their combination in predicting different recurrence sites. For some patients whose tumor markers were negative, we explored whether NLR, HALP, and PLR could be used as indicators in predicting prognosis in ICC patients.

Comment 5: The manuscript is too long and should be shortened.

Reply: thank you for your comments of our study, we have made substantial changes accordingly to make it more concise. The revised manuscript with changes accordingly is uploaded.

Reviewer 2 Report

In this retrospective study, the authors analyze the predictive value of several tumours markers, as well the  

Predictive value of HALP, NLR and PLR scores. To my understanding, they have not reached a clear conclusion, which will make the management of intrahepatic cholangiocarcinoma more effective.

Furthermore, they do not comment on the prognostic value of positive lymph nodes in the surgical specimen.

Also, in the era of genetic analysis, they should review the literature on the role of the preoperative genetic profile in pretreatment therapeutic decision-making. 

Author Response

Comment 1: In this retrospective study, the authors analyze the predictive value of several tumours markers, as well the Predictive value of HALP, NLR and PLR scores. To my understanding, they have not reached a clear conclusion, which will make the management of intrahepatic cholangiocarcinoma more effective.

Reply: Thank you very much for your comments and professional advice. In this research, we found that only CA19-9 in liver tumor markers was related to the RFS in ICC patients. However, the modes of recurrence in ICC patients often conclude intrahepatic recurrence, lymph node metastasis, and distant metastasis, and the treatment modalities differs from one another. So, we carried out this research to investigate whether these single liver tumor marker or their combination could have the prognostic prediction ability in these different modes of recurrence. Regrettably, CA19-9 showed its value in predicting distant metastasis, but not intrahepatic recurrence or lymph node metastasis. So, we focused on the preoperative HALP score, NLR, and PLR, which had been reported to have predictive value in breast cancer (PMID: 35702128), small-cell lung cancer (PMID: 33004760), myeloma (PMID: 36053932), and so on. Surprisingly, HALP had a nice predictive ability in predicting intrahepatic recurrence and NLR had a nice predictive value in predicting lymph node metastasis recurrence. And these are simple, cost-effective, easily accessible indexes.

Comment 2: Furthermore, they do not comment on the prognostic value of positive lymph nodes in the surgical specimen.

Reply: Thank you for your suggestions for our research. We added a survival analysis based on the predictive value of lymph nodes (Figure supplement 2F, 3F), and Patients with positive lymph nodes showed poorer prognosis than those with negative lymph nodes.

Comment 3: Also, in the era of genetic analysis, they should review the literature on the role of the preoperative genetic profile in pretreatment therapeutic decision-making.

Reply: Thank you for your suggestions for our research. Genetic analysis indeed could provide much useful information for us in personalized treatment. However, in spite of improving the accuracy of prediction, the high costs of these emerging technologies make their use limited. Therefore, we aim this study to develop simple, inexpensive, reliable indicators for predicting prognosis in ICC patients. With the emerging studies relating the next-generation sequencing (NGS) results and treatment options, adding genetic profile into consideration is definately the future trend. Genetic analysis also provides ideas for our future research, and we look forword to address this in future research.

Reviewer 3 Report

Dear Editor,

I reviewed the manuscript by Zhanget al., entitled "Liver Tumor Markers, HALP Score, and NLR: Simple, 2 Cost-effective, Easily Accessible Indexes for Predicting 3 Prognosis in ICC Patients after Surgery" this manuscript may provide a new information for the reader of the journal. However, The authors need to improve the qality of by reducing the figures to make it interesting for the réader. So much data will not helpful to understand The main context of the mauscript.

Many thanks

Author Response

Comment 1: I reviewed the manuscript by Zhanget al., entitled "Liver Tumor Markers, HALP Score, and NLR: Simple, Cost-effective, Easily Accessible Indexes for Predicting Prognosis in ICC Patients after Surgery" this manuscript may provide a new information for the reader of the journal. However,  The authors need to improve the qality of by reducing the figures to make it interesting for the réader. So much data will not helpful to understand The main context of the mauscript.

Reply: Thank you for your approval of our study, and this is exactly one of our primary concerns. According to your valuable suggestions, we reduced the length of the article and put some figures in the supplemental materials to make it more clear and more concise. The revised manuscript with changes accordingly is uploaded.

Round 2

Reviewer 1 Report

The new version of the manuscript is more readable and concise. However the results are still poor, without a significant message for the clinicians.

Author Response

We sincerely appreciate Reviewer 1 for the delicate revision of our manuscript. Regarding these valuable suggestions, we made substantial changes to improve the quality of our study. This is the first study to investigate liver tumor markers (AFP, PIVKA-II, CA19-9, and CEA) as a prognostic biomarker in ICC, and there may indeed be some shortcomings. The levels of PIVKA-II were not associated with the prognosis outcomes of ICC patients, but an increased level was related to more advanced BCLC staging and AJCC staging, and fewer tumor numbers (p < 0.05). The reasons for this phenomenon may be related to the cut-off value of PIVKA-II. The cut-off value adopted in the study was referenced to another study investigating the value of PIVKA-II in a large HCC cohort (Pan YX, Sun XQ, Hu ZL, et al. Prognostic Values of Alpha-Fetoprotein and Des-Gamma-Carboxyprothrombin in Hepatocellular Carcinoma in China: An Analysis of 4792 Patients. J Hepatocell Carcinoma. 2021). Surely, the best cut-off value of PIVKA-II in ICC patients would have to be done in a larger prospective study.

In addition, the modes of recurrence in ICC patients often conclude intrahepatic recurrence, lymph node metastasis, and distant metastasis, and the treatment method is different after relapse. In our study, the HALP score could have the potential ability in predicting intrahepatic recurrence, NLR could have a nice predictive value in predicting lymph node metastasis recurrence after surgery, and CA19-9 levels could have a nice ability in predicting distant metastasis. We hope these could provide useful information for clinicians before surgery.

We will be happy to edit the text further, based on helpful comments from yours.

Reviewer 2 Report

The Authors have responded satisfactorily to my comments. 

I would suggest the reply in comment 3 to be included in the discussion section. 

Author Response

Thank you for your suggestions for our research. We have added the comments on genetic analysis in the discussion section according to your valuable suggestion, and the revised part was marked in red.